# microRNAs in Human Adipose Tissue Physiology and Dysfunction

**DOI:** 10.3390/cells10123342

**Published:** 2021-11-28

**Authors:** Alina Kurylowicz

**Affiliations:** 1Department of Human Epigenetics, Mossakowski Medical Research Centre, Polish Academy of Sciences, 02-106 Warsaw, Poland; 2Department of General Medicine and Gerontocardiology, Medical Centre of Postgraduate Education, 00-401 Warsaw, Poland

**Keywords:** microRNA (miRNA), adipose tissue, obesity

## Abstract

In recent years, there has been a large amount of evidence on the role of microRNA (miRNA) in regulating adipose tissue physiology. Indeed, miRNAs control critical steps in adipocyte differentiation, proliferation and browning, as well as lipolysis, lipogenesis and adipokine secretion. Overnutrition leads to a significant change in the adipocyte miRNOME, resulting in adipose tissue dysfunction. Moreover, via secreted mediators, dysfunctional adipocytes may impair the function of other organs and tissues. However, given their potential to control cell and whole-body energy expenditure, miRNAs also represent critical therapeutic targets for treating obesity and related metabolic complications. This review attempts to integrate present concepts on the role miRNAs play in adipose tissue physiology and obesity-related dysfunction and data from pre-clinical and clinical studies on the diagnostic or therapeutic potential of miRNA in obesity and its related complications.

## 1. Introduction

According to worldwide statistics, the number of obese individuals has tripled in the last 30 years, and it is estimated that in 2025, they will constitute approximately 15% of the world adult population. However, it is not obesity itself but obesity-related complications that affect virtually all organs in the body and significantly deteriorate quality of life, constituting a severe social and economic problem [1]. The development of obesity-related complications is related to the dysfunction of adipose tissue. The excessive accumulation of lipids changes the adipocyte metabolism, leading, among other things, to dysfunction of the mitochondria and the associated endoplasmic reticulum stress [2]. These phenomena influence adipocyte transcriptional activity and thus the profile of substances secreted by adipose tissue (adipokines), which affects the functioning of tissues and organs throughout the body in an endocrine manner. While the effects of obesity-related adipose tissue dysfunction are already known, the knowledge of the underlying mechanisms is still insufficient.

In recent years, there has been much interest in the potential role of microRNA (miRNA) in regulating gene expression in adipose tissue [3,4]. miRNAs are single-stranded noncoding RNAs of 17–25 nucleotides in length (on average 22). Although miRNAs do not have open reading frames (they do not encode proteins), they can perform regulatory functions in the cell. It has been shown, for example, that miRNAs located in the cell nucleus can regulate gene transcription both directly—acting as transcriptional cofactors and participating in the mRNA maturation process—and indirectly—by influencing the chromatin structure by participating in histone methylation and acetylation [5]. 

In vitro studies and animal models of obesity have shown that some miRNAs regulate adipogenesis and browning of adipose tissue [4]. Moreover, the targeted overexpression or ablation of particular miRNAs in white adipose tissue results in enhanced mitochondrial biogenesis and a subsequent decrease in insulin resistance, an improvement in the plasma lipid profile and a reduction in fatty liver markers in mice with diet-induced obesity (DIO) [6,7]. There is evidence that miRNAs can also regulate the function of adipocytes in humans, and obesity may lead to a change in the miRNA profile in adipose tissue [8,9]. Since the manipulation of these miRNAs has been shown to affect whole-body energy expenditure, glucose tolerance and insulin sensitivity in vivo, they represent potentially critical therapeutic targets for treating obesity and related metabolic complications.

One should also remember that miRNAs secreted by adipose tissue not only act locally but also reach the bloodstream, where they constitute 80% of circulating miRNAs. Subsequently, miRNA secreted by dysfunctional adipocytes in the course of obesity can affect the function of distant organs and tissues, contributing to the development of obesity-associated complications [3]. This review attempts to integrate present concepts on the role miRNAs play in adipose tissue physiology and obesity-related dysfunction, as well as data from pre-clinical and clinical studies on the diagnostic or therapeutic potential of miRNA in obesity and related complications. It begins with a brief overview of the mechanisms by which miRNAs may regulate gene expression to provide a background for discussion of their role in adipose tissue development and function. Then, the role of miRNA in white and brown adipogenesis and browning of white adipose tissue is presented. Finally, the role of various miRNAs in regulating the physiological functions of adipose tissue (including lipolysis/lipogenesis and adipokine secretion) and pathological states (inflammatory activity and insulin resistance) is discussed.

## 2. miRNAs as Regulators of Gene Expression

Since their discovery in 1993, there has been growing evidence on the role that miRNAs play in various biological processes, both in physiology and disease. In addition to their engagement in the local regulation of gene expression, miRNAs are secreted into extracellular fluids and take part in intracellular and intraorgan communication [5]. The process of miRNA biogenesis is not a subject of this review and is described in detail elsewhere [10]. Briefly, miRNAs are transcribed from DNA sequences into primary miRNAs (pri-miRNAs) and subsequently processed into precursor miRNAs (pre-miRNAs) and, finally, mature miRNAs. Some miRNAs are intragenic and transcripted into pri-miRNAs from introns of protein-encoding genes, while others are intergenic, and their promoters regulate their transcription independently of the host genes [11].

Interestingly, the primary transcripts of miRNAs may encode small peptides, miPEPs, which can specifically activate the transcription of their miRNA in a positive loop [12]. During a so-called canonical biogenesis pathway, pri-miRNAs are processed by the microprocessor complex, consisting of an RNA-binding protein, namely, DiGeorge Syndrome Critical Region 8 (DGCR8), and Drosha (a ribonuclease III enzyme), into pre-miRNAs. Subsequently, an exportin 5 (XPO5)/RanGTP complex exports pre-miRNA from the nucleus to the cytoplasm, processed by the RNase III endonuclease Dicer. After the removal of the terminal loop, a mature miRNA duplex is formed. The 3p miRNA strand arises from the 3′ end of the pre-miRNA hairpin, while the 5p originates from the 5′ end of the mature miRNA duplex. Then, both miRNA strands can be loaded into the Argonaute (AGO) family of proteins (AGO1-4 in humans) in an ATP-dependent manner, which determines their further action [13]. However, several miRNAs are processed in noncanonical pathways, generally categorized as Drosha/DGCR8-independent and Dicer-independent [10]. 

In most cases, miRNAs induce translational repression and mRNA deadenylation and decapping by binding to a specific sequence in the 3′ untranslated regions (UTRs) of their target mRNAs. However, the silencing effects on gene expression can also be exerted by miRNA binding to the 5′ UTR and coding sequence. In turn, binding to the sequences located within promoter regions has been reported to induce transcription under certain conditions [14].

The miRNA-mediated gene silencing occurs through the minimal miRNA-induced silencing complex (miRISC) consisting of a guide miRNA strand and AGO proteins. miRISC target specificity depends on its interactions with complementary sequences on target mRNA (miRNA response elements—MREs). The degree of miRNA/MRE complementarity determines whether the interaction results in AGO2-mediated target mRNA silencing or miRISC-dependent translational inhibition followed by the degradation of the target mRNA. In the case of complete complementarity, AGO2 exerts its endonuclease activity, resulting in the cleavage of the target mRNA and degradation of the miRNA [15].

However, when the miRNA/MRE interaction is not entirely complementary, the sequence mismatches prevent AGO2 endonuclease activity. As in the case of other AGO proteins that do not have endonucleolytic properties, it acts as a mediator of RNA interference [16]. The miRNA-mediated translational inhibition via the miRISC complex starts with the recruitment of the GW182 family of proteins, which provides the scaffolding necessary to recruit other effector proteins after miRNA/mRNA interaction. These proteins include the poly(A)-deadenylase complexes PAN2-PAN3 and CCR4-NOT responsible for the target mRNA poly(A)-deadenylation. Then, decapping protein 2 (DCP2), alongside its associated proteins, initiates mRNA decapping and subsequent 5′-3′degradation via exoribonuclease 1 (XRN1) [17].

While most miRNA/mRNA actions result in gene silencing, miRNA can also activate translation under certain conditions. This takes place, for instance, in the case of binding to the 5′ UTR sequence of mRNAs encoding ribosomal proteins during amino acid starvation [18]. Another example includes the interaction of miRNAs with AGO2 and Fragile-x-mental retardation-related protein 1 (FXR1, instead of GW182), resulting in the activation of translation during cell cycle arrest. However, in proliferating cells, the same interaction inhibits translation [19,20]. Interestingly, when one member of an miRNA family has a mutation in its canonical cleavage site, it can still interact with the target mRNA and protect it against silencing by other family members. In this case, one miRNA competes with the other members of its family to prevent their activity. Moreover, miRNA can be neutralized by long RNA targets containing a complementary sequence with a mismatched loop at the cleavage site that sequesters the miRNA and, in this way, prevents its activity [21].

miRNAs also participate in the regulation of transcriptional and post-transcriptional gene activity in the nucleus. This is possible since both AGO2 and Exportin-1/Impotin-8 proteins, due to the interaction with Trinucleotide Repeat Containing Adaptor 6A (TNRC6A) containing a nuclear localization and export signal, can shuttle between the nucleus and cytoplasm [22]. The mechanisms of miRNA action in the nucleus are still to be elucidated. However, there are data reporting a direct interaction between miRNA and promoters of target genes [23], and the ability of miRNA to upregulate gene activity by promoting a transcriptionally active chromatin state or to participate in alternative splicing [22,24]. It is also suggested that miRISC is involved in regulating DNA and histone methylation status and, in this way, in genomic remodeling [5]. 

Discovery of the mechanisms by which miRNAs can influence gene expression has led to an understanding of their role in regulating the development and function of many organs and tissues, including adipose tissue.

## 3. Role of microRNAs in Adipogenesis

In recent years, it has been found that miRNAs play a central role in regulating both white and brown adipocyte proliferation and differentiation, as well as in adipose tissue browning. A detailed description of all the processes by which miRNAs are involved in adipogenesis is beyond the scope of this review. Here, particular attention will be paid to those stages of adipocyte proliferation and differentiation that are disturbed in dysfunctional adipose tissue in the course of obesity to underline the role of miRNAs in their regulation.

### 3.1. miRNA in the Regulation of White Adipogenesis

White adipose tissue (WAT) not only constitutes a significant energy depot in the human body but also secretes numerous mediators that can affect the function of distant organs. Complex interactions between different signaling pathways and transcription factors determine the process of multipotent mesenchymal stem cell (MSC) differentiation to white preadipocytes followed by growth arrest induced by contact inhibition. Next, adipogenic stimuli promote cell cycle reentry and synchronous cell division. This process depends on the induction of transcription factors, which are members of the CCAAT/enhancer-binding protein (C/EBP) family. Subsequently, C/EBPβ activates the transcription of the major transcriptional inducers of adipogenic gene expression: *C/EBPA* and peroxisome-proliferator-activated receptor γ (*PPARG*), as well as several other transcription factors responsible for the terminal differentiation of preadipocytes into lipid-storing mature adipocytes [25]. Two key signaling pathways involved in MSC differentiation toward adipocytes are the Wingless (Wnt)/β-catenin and the transforming growth factor β (TGFβ)/bone morphogenetic proteins (BMP)/Smad pathways. Activated Wnt/β-catenin signaling stimulates osteogenic gene expression; however, its inhibition induces adipogenic differentiation via the induction of adipogenic-related genes, including the abovementioned *PPARG* and *C/EBPA*. The interaction between β-catenin and PPARγ results in β-catenin degradation with the subsequent inhibition of Wnt/β-catenin signaling and bone formation, promoting adipogenesis. Similarly, BMP/Smad signaling favors adipogenesis by increasing the expression of *PPARG*, while the TGFβ/Smad pathway exerts inhibitory effects on adipogenesis [26,27]. The proper regulation of adipocyte development and turnover assures adipose tissue homeostasis, which is seriously disturbed in the course of obesity.

The inhibition of enzymes involved in miRNA biogenesis, such as Drosha and Dicer, depresses the differentiation of human MSCs into adipocytes, which supports a role for miRNAs in adipocyte development [28]. Subsequently, several miRNAs have been found to be regulators of human adipocyte differentiation [4]. WAT undergoes dynamic changes to adapt to the body’s energy balance to maintain its energy storage role. In excess energy intake, WAT augments its capacity to store energy by increasing lipid accumulation and differentiation of preadipocytes to mature adipocytes. The remodeling of WAT in response to the excess of nutrients is accompanied by changes in the WAT miRNOME, with several miRNAs being up- and downregulated [29]. Moreover, several adipocyte-selective miRNAs implicated in adipocyte proliferation and differentiation in normal-weight individuals are differentially expressed in adipose tissues of obese subjects [8,9,30].

miRNAs are implicated in the regulation of the critical signaling pathways related to adipogenesis. For instance, miR-9-5p, by targeting the 3’UTR of Wnt3a (a Wnt ligand) and reducing its expression, inhibits Wnt/β-catenin signaling and promotes the differentiation of rat MSCs toward adipocytes [31]. The expression of miR-9-5p is significantly increased in visceral adipose tissue (VAT) compared to subcutaneous adipose tissue (SAT) in obese patients, suggesting that the upregulation of this miRNA can be involved in the pathogenesis of obesity in humans [9]. High serum miR-9 levels are also considered as a marker of poor prognosis in diabetic nephropathy [32]. A similar effect exerts miR-210, which targets the T cell-specific transcription factor 7-like 2 (TCF7L2) responsible for triggering the downstream responsive genes of the Wnt pathway, as previously shown in 3T3-L1 murine preadipocytes [33]. This miRNA was, in turn, upregulated in the SAT of obese patients, compared to the SAT of normal-weight individuals, while weight loss led to a significant decrease in its expression [8,9]. Moreover, the expression of miR-210 was upregulated in the SAT of obese individuals with normal glucose tolerance, compared to those diagnosed with type 2 diabetes mellitus (T2DM), suggesting its protective role in the development of obesity-related adipose tissue dysfunction [34]. However, in patients with T2DM, serum miR-210 concentration may serve as a diagnostic biomarker of diabetic retinopathy patients and may have the ability to predict disease development and severity [35].

In turn, miR-21 was found to positively regulate adipogenesis in human adipose-derived stromal cells (hADSCs) by binding and neutralizing TGFβ1—an inhibitor of adipogenesis [36]. The upregulation of miR-21 in the adipose tissue of obese subjects has been consistently reported in several studies [9,37,38], while its serum levels correlate negatively with body mass index, waist circumference and insulin levels [39]. Another miRNA targeting TGFβ/Smad is miR-199a-5p, which promotes the adipogenic differentiation of human bone marrow stromal cells (BMSCs) [40]. Obesity is associated with increased miR-199a-5p levels in VAT and in sera [34,41]. However, metabolically healthy obese individuals have lower miR-199a-5p expression in SAT compared to obese patients diagnosed with T2DM [8]. The regulation of adipogenesis can also be obtained by miRNAs acting on TGFβ/Smad-independent pathways, as shown in the case of miR-143 targeting extracellular-signal-regulated kinase 5 (ERK5) [42]. The regulatory role of this miRNA during adipogenesis depends on the differentiation stage that it acts on. If miR-143 is overexpressed during the clonal expansion stage, it inhibits the adipogenic differentiation of adipose tissue-derived stromal cells (ADSC). On the contrary, miR-143 overexpression during the growth arrest stage or terminal differentiation stage promotes adipocyte differentiation [43]. The expression of this miRNA is upregulated in the sera of obese individuals and decreased in the adipose tissue of previously obese patients after successful weight loss [44]. 

Several other miRNAs have been shown to inhibit MSC differentiation towards osteoblasts and, therefore, to promote adipogenesis. This takes place in the case of miR-204 and miR-637, which target members of the Wnt/β-catenin signaling pathway: runt-related transcription factor 2 (RUNX2) and osterix (OSX), respectively [45,46]. The increased expression of miR-204 characterizes VAT in human obesity and impairs mitochondrial biogenesis and the development of brown adipose tissue (BAT) in rodents [9,30]. Obesity-related changes in miR-637 expression in human adipose tissue have not been reported to date; however, its serum levels increase during dietary weight loss intervention [47].

In turn, miR-27b and miR-130a, by interfering with PPARγ and miR-31 by targeting C/EBPα, were found to favor the osteogenic differentiation of MSCs [48,49,50]. Moreover, miR-130a can suppress MSC differentiation towards adipocytes via interfering with adenomatosis polyposis coli downregulated 1 (*APCDD1*), encoding an inhibitor of the Wnt signaling pathway [51]. Consistently, decreased levels of this miRNA accompany human preadipocyte differentiation [8,38,52]. However, while the decreased expression of miR-27b was found in the SAT of obese patients with T2DM and VAT, and individuals with non-alcoholic steatohepatitis (NASH), the SAT of metabolically healthy obese individuals was characterized by increased levels of miR-27b, and weight loss did not influence its expression [9,30,37,53]. In turn, obese patients with polycystic ovary syndrome (PCOS) tended to exhibit decreased serum miR-27b levels [54]. Data on obesity-induced changes in miR-130a are inconsistent; in some studies, obesity was associated with increased miR-130a levels in SAT and in sera, while in others, it was associated with its downregulation. However, it should be noted that while Nardelli et al. measured the expression of both miR-130a isoforms (3p and 5p), in the study by Wang et al., only the miR-130a-5p level was assessed [30,37,55]. In turn, miR-31-5p was found to be upregulated in the VAT compared to the SAT of obese adult individuals and in the sera of obese adolescents [9,56]. 

miR-181a is an example of another miRNA regulating adipogenesis by targeting PPARγ; however, miR-181a suppression decreased the expression of PPARγ in porcine primary preadipocytes [57]. The role of this miRNA in the pathogenesis of obesity and related complications in humans is less clear. Ortega et al. found that increased miR-181a expression is a hallmark of preadipocytes originating from the SAT of obese subjects, compared to normal-weight individuals, while Kloting et al. observed its upregulated levels in the SAT of T2DM patients compared to obese subjects without metabolic complications of obesity [8,34]. Moreover, miR-181a expression was significantly elevated in the serum of patients with non-alcoholic fatty liver disease (NAFLD), suggesting it can serve as a disease marker [58]. However, in other studies, miR-181a expression was downregulated in the SAT of metabolically healthy obese patients and VAT of obese individuals with NASH [53,59].

In addition to the direct interaction with PPARγ mRNA, miRNA can regulate adipogenesis by targeting proteins involved in the regulation of PPARγ activity. For instance, miR-146b has been found to promote preadipocyte differentiation via interaction with Sirt1, the NAD-dependent deacetylase, known as a PPARγ inhibitor [60]. Human obesity is associated with the upregulated expression of miR-146b in SAT and sera, which significantly decreases after weight loss [9,61,62]. Interestingly, a high miR-146b serum concentration is a T2DM predictor in obese adolescents, while lower levels are observed exclusively in the VAT of patients with NASH and pericellular fibrosis but are not changed between NASH and non-NASH NAFLD patients [53,62].

The decreased expression of *SIRT1* correlates negatively with the expression of several other miRNAs in the adipose tissues of obese patients, including those involved in pro-inflammatory responses (miR-22-3p), the inhibition of adipose tissue browning (miR-34a-5p) and the activation of white adipogenesis (miR-181a-3p), suggesting that the interaction between miRNAs and SIRT1 constitutes a critical regulatory mechanism in adipocyte homeostasis [57,59,63,64].

Given that a single miRNA targets several mRNAs, its regulatory influence on adipocyte differentiation can be exerted by triggering different cellular pathways. For instance, in different experimental conditions, miR-103 was found to promote adipogenesis via (i) targeting retinoic acid-induced protein 14 (RAI14) in the early stages of adipogenesis; (ii) activation of the protein kinase B/mammalian target of a rapamycin signaling pathway (AKT/mTOR pathway); and (iii) reversing the anti-adipogenic effects of myocyte enhancer factor 2D (MEFD2), which is a transcription factor that negatively regulates preadipocyte differentiation by downregulating the expression of multiple adipocyte markers (e.g., PPARγ and C/EBPα) [65,66]. However, in humans, obesity seems to have little influence on miR-103 expression in adipose tissue, and thus its isoform, miR-103a-3p, is even recommended as a reference for the analysis of miRNA expression in adipocytes [67]. On the contrary, miR-103, together with miR-107, can promote endoplasmic reticulum stress-mediated apoptosis in murine preadipocytes by targeting the Wnt3a/β-catenin/activating transcription factor 6 (ATF6) signaling pathway [68]. The expression of miR-107 was downregulated in the SAT of obese individuals compared to normal-weight subjects and in the VAT of those diagnosed with NASH [37,53,69]. Moreover, miR-107 SAT levels decreased after bariatric surgery, suggesting persistent, obesity-related adipose tissue dysfunction [9]. In turn, elevated miR-103/miR-107 serum levels are a predictor of insulin resistance in obese adolescents [70].

For many years, BAT has been thought to play a marginal role in adult energy homeostasis. However, recent research has increased our understanding of the mechanisms involved in the development and activation of brown adipocytes and their contribution to metabolic health. It has been revealed that miRNAs also play a significant role in regulating these processes.

### 3.2. miRNA in Regulation of Brown Adipogenesis and Thermogenesis 

The proper development and activity of BAT enable higher metabolic rates and can protect against the development of obesity. The brown adipocytes present in the human body can be of distinct developmental origins. The classical or constitutive brown adipose tissue (cBAT) expands during embryogenesis, and recruitable BAT (rBAT, alternatively called beige or brite) emerges postnatally within WAT in the adipose tissue browning process (described in the following section). The activation of adaptive thermogenesis to maintain the normal body temperature is the primary role of cBAT. This is feasible due to the high content of mitochondrial uncoupling proteins (UCPs) responsible for the uncoupling of electron transport from the production of chemical energy in the form of adenosine triphosphate (ATP). The change in the balance of electrons and protons across the mitochondrial membrane leads to energy loss as heat is essential to preserve the normal body temperature [71].

In addition to the general regulators of adipogenesis common with white adipocyte development, the expression of thermogenic genes in brown and beige adipocytes requires additional transcriptional factors, including peroxisome PPARγ coactivator 1 α (PGC1α), PR domain containing 16 (PRDM16) and forkhead box C2 (FOXC2) (reviewed in [72]). Several miRNAs have been reported to regulate brown adipogenesis and, subsequently, the BAT thermogenic program. Therefore, the up- or downregulation of these miRNAs may influence the effectiveness of thermogenesis, which affects whole-body energy expenditure and glucose uptake, and insulin sensitivity [73]. Since miRNAs involved in brown adipogenesis also frequently participate in regulation of thermogenic pathways, they will be discussed together in the following sections.

#### 3.2.1. miRNA Involved in Constitutive Brown Adipose Tissue (cBAT) Development

Two miRNAs essential for brown adipogenesis are miR-193a/b and miR-365, which form a cluster located on chromosome 16. This cluster is positively regulated by PRDM16 and targets genes involved in the inhibition of adipogenesis (e.g., runt-related transcription factor 1 Partner Transcriptional Co-Repressor 1, RUNX1T1) and activation of myogenesis (e.g., surface protein Cdo and insulin-like growth factor binding protein 5, IGFBP5). The inhibition of the miR-193a/b and miR-365 cluster in primary murine adipocytes leads to the suppression of thermogenic genes (Ucp1, Pgc1α, Pparα and Prdm16) but also genes that are crucial for both brown and white adipocyte development and function (e.g., Pparγ) [74]. However, the crucial role of the miR-193a/b and miR-365 cluster in BAT development and function has not been confirmed by animal studies since, for instance, miR-193b-mutant mice have normal cold tolerance and unaffected expression of essential thermogenic genes in BAT (including Prdm16 and Ucp1) [75]. In humans, obesity is associated with the decreased expression of miR-193a/b and miR-365 in the SAT and VAT of obese subjects with co-existing NASH [8,30,53] which may increase after surgically induced weight loss [9]. In turn, serum miR-193b levels correlate negatively with body mass index but are increased in patients with prediabetes and normalize upon regular exercise [76,77]. However, whether these alternations in the expression of the miR-193a/b and miR-365 cluster translate into changes in brown/beige adipocyte counts, and thus the metabolic activity of the adipose tissue, remains unknown. 

The seed sequence of miR-193b resembles miR-328, which acts as a positive regulator of brown adipogenesis. When overexpressed, miR-328 stimulates the BAT differentiation of brown adipocytes by suppressing the expression of muscle lineage regulators (e.g., β-secretase—BACE1). In turn, miR-328 silencing leads to a significant downregulation of thermogenic genes in the brown adipose tissue of aging and diet-induced obese mice [78]. Interestingly, miR-328 was observed to play a critical role in the regulation of cystathionine β-synthase activity, which translates to improved liver protein metabolism efficiency in mice with DIO, while its decreased expression was detected in the VAT of obese patients diagnosed with NASH [53,79]. Low serum miR-328 levels are also a hallmark of obesity in children [80]. 

Another example of a positive regulator of adipogenesis in cBAT is miR-378. Its overexpression leads to BAT expansion, resulting in increased energy expenditure, and protects against genetically determined (in ob/ob mice) and diet-induced obesity [81]. The molecular mechanism underlying this phenomenon involves the regulation of cAMP turnover in brown adipocytes due to the downregulation of PDE1B phosphodiesterase. Levels of this miRNA increase in the course of normal adipogenesis in humans [8]. However, its expression is decreased in the SAT of obese individuals and VAT of obese subjects diagnosed with NASH [30,53]. In turn, serum miR-378 concentrations can serve as a marker of insulin resistance [82]. 

miR-182 and miR-203 also act as activators of brown adipogenesis. Their inhibition causes the downregulation of brown adipocyte- and mitochondrial-specific genes (including Ucp1, Pgc1α, Pparα, Cox7 and Cox8) in mice, but does not influence the expression of the common adipogenic markers (e.g., Pparγ) [83]. Nevertheless, in humans, obesity is associated with higher miR-182 and miR-203 levels in VAT than SAT, while weight loss leads to the significant downregulation of miR-182 in SAT [9,34]. Given these discrepancies, further studies are required to clarify the role of miR-182 and miR-203 in the regulation of BAT development and function in humans.

However, specific miRNAs have been reported to act as negative regulators of BAT differentiation; two examples of such miRNAs are miR-106b and miR-93 (belonging to the miR-17 family) that form a cluster. Their knockdown induces BAT-specific genes (e.g., Ucp1, Prmd16, Ppara and Pgc1a) in murine adipocytes, while their ectopic expression correlates with lower Ucp1 mRNA levels. Higher expression of the miR-106b/miR-93 cluster in the adipose tissue of animals on a high-fat diet (HFD) suggests that diet-induced obesity leads to the depletion of BAT and impairment of its thermogenic activity, which has also been found in human adipose tissue [84,85]. In obese humans, increased expression of both miR-106b isoforms (5p and 3p), as well as miR-93-5p, in SAT is reduced after weight loss [9]; however, this phenomenon does not occur with miR-93-3p, in which SAT levels are depleted in the course of obesity [30].

Examples of other miRNAs that inhibit differentiation toward brown adipocytes include miR-34a, miR-27b, miR-133, miR-155, miR-32, miR-455 and miR-30b/c. However, since they are also involved in regulating adipose tissue browning, their role in adipocyte differentiation and function is discussed in the following section.

#### 3.2.2. miRNA Involvement in Recruitable Brown Adipose Tissue (rBAT) Development

The developmental origin of rBAT is distinct from cBAT. Beige adipocytes differentiate either from adipocyte progenitor cells or from mature white adipocytes in the process called transdifferentiation. The stimulation of adrenergic receptors β3 (leading to the induction of PPARγ and PGC1α), as well as the activation of SIRT1 (responsible for the activation of PPARγ and PRDM16), is crucial for the initiation of the brown fat-specific program and thermogenic pathways (reviewed in [72]). Since miRNAs regulate the action of the abovementioned transcription factors, they are also involved in regulating adipose tissue browning.

For instance, miR-34a, by disrupting fibroblast growth factor 21 (FGF21) and SIRT1 signaling, prevents the deacetylation of PGC1α and subsequently inhibits the browning transcriptional program in 3T3-L1 adipocytes [64]. The high expression of miR-34a correlates negatively with SIRT1 mRNA levels in human adipose tissue and is associated with obesity in mice and humans [8,37,59,64]. Increased miR-34a serum levels in obese individuals predict insulin resistance [82]. Subsequently, the downregulation of miR-34a in BALB/c mice with DIO increased expression of markers of beige fat (including Ucp1, Prmd16 and Pgc1a) in all types of WATs and promoted BAT expansion, which resulted in reduced adiposity [64].

miR-27b is another example of a brown and beige adipogenesis inhibitor. Its overexpression in human multipotent adipose-derived stem (hMADS) cells not only blunted the induction of PPARγ and C/EBPα during the early stages of adipogenesis but also, by targeting prohibitin (PHB, an essential protein for mitochondria homeostasis), impaired mitochondrial biogenesis, structure integrity and activity, which was accompanied by excessive reactive oxygen species production [86,87]. The list of the miR-27b targets includes typical BAT markers, such as UCP1, PRMD16, PPARα or PGC1α, and its downregulation increases WAT browning in DIO mice, leading to weight loss and improved insulin sensitivity, while overexpression inhibits the browning ability of white adipocytes [88]. In turn, stimuli inducing adipose tissue browning, such as cold exposure and β-adrenergic activation, lead to the significant downregulation of miR-27b in murine adipose tissue [89]. However, the role of miR-27b in the development of human obesity has not yet been clarified, since excess adiposity is associated with its downregulation in human SAT [30,37], while NASH is also associated with its downregulation in VAT [53].

Cold exposure and β-adrenergic activation also lead to the depletion of the miR-133 level, resulting in the upregulation of PRMD16 and thermogenic genes (including PPARG, PPARA and UCP1) in BAT and the promotion of white adipocyte browning in SAT. Accordingly, mice with miR-133 knockout have elevated expression of genes associated with adipocyte browning and thermogenesis in SAT, which is accompanied by increased insulin sensitivity and glucose tolerance compared to wild-type animals [90]. On the contrary, the overexpression of miR-133 in brown adipogenic conditions prevents the differentiation to brown adipocytes in both BAT and SAT precursors [91]. Both miR-133a and miR-133b are upregulated in VAT in the course of obesity in humans [9].

The regulatory pathways involved in adipose tissue browning are complex and frequently contain mutual feedback loops. For instance, C/EBPβ, which, in addition to its general role in adipogenesis, is a crucial transcription factor for brown and beige adipocyte differentiation, negatively regulates the expression of miR-155, which, in turn, downregulates C/EBPβ. miR-155 inhibition enhances brown adipocyte differentiation and induces browning in white adipocytes. Its knockout results in increased WAT browning and mitochondrial activity in brown adipocytes, while transgenic overexpression causes a reduction in BAT mass and impairment of brown adipocyte thermogenic function in mice [92]. Importantly, miR-155 knockout mice are characterized by increased WAT browning and resistance to HFD-induced obesity, increased insulin sensitivity and a decreased adipose tissue concentration of inflammatory parameters [6]. Surprisingly, miR-155 expression was downregulated in the SAT of diabetic obese patients compared to obese, metabolically healthy individuals, and its serum levels correlated negatively with body mass index and cholesterol levels in non-diabetic patients [34,93].

While the abovementioned miRNAs act as negative regulators of WAT browning and BAT thermogenic activity, several miRNAs were found to enhance these processes. For instance, cold exposure leads to the BAT-selective upregulation of miR-32 in C57BL6/J mice. This miRNA targets Tob1 (transducer of erbB2 1, a p38/AMP-activated protein kinase (AMPK) signaling repressor), resulting in the upregulation of p38 mitogen-activated protein kinase signaling and increased expression of FGF21. FGF21, by enhancing PGC1α and UCP1, acts as a positive regulator of brown adipocyte differentiation and thermogenesis. Moreover, transgenic mice with miR-32 overexpression in BAT are characterized by increased FGF21 serum levels and enhanced WAT browning. BAT-specific inhibition of miR-32 reduces its mass, and thermogenic activity decreases FGF21 expression in BAT, resulting in impaired beige cell recruitment in WAT [7]. On the contrary, in humans, weight loss is accompanied by reduced miR-32 expression in SAT, while obesity seems not to influence miR-32 serum levels [9,82].

miR-455 is another example of a positive regulator of BAT function and WAT browning. By targeting upstream regulators (including RUNX1T1, Necdin and hypoxia inducible factor 1 subunit alpha inhibitor—HIF1AN), miR-455 enhances AMPKa1 activity, leading to the increased expression of key genes involved in the regulation of adipogenesis (e.g., Pparγ and Cebpα) and thermogenic activity of brown adipocytes (e.g., Ucp1, Pgc1a, Prdm16 and Cidea) in committed brown and white preadipocytes and in non-committed multipotent progenitor cells in vitro. WAT-specific overexpression of miR-455 leads to its marked browning and increased thermogenic capacity, which translates to an improvement in cold resistance, as well as several metabolic parameters, including glucose tolerance, insulin sensitivity and lipidemic profiles, despite an HFD being fed to FAT455 mice (mice overexpressing miR-455 in adipose tissue using an aP2 promoter-driven transgene). In turn, mice with miRNA-455 knockdown have decreased BAT and WAT mass and suppressed expression of key thermogenic and adipogenic markers (e.g., Ucp1, Pgc1α, Pparγ and Cebpα) [94]. However, the positive correlation between miR-455 and Ucp1 expression does not result from their direct interaction but from the downregulation of several other mRNA-encoding negative regulators of adipose tissue browning and thermogenesis [95,96]. Nevertheless, human adipocytes derived from the SAT of obese individuals are characterized by higher miR-455 levels than lean subjects’ cells [8].

While miR-30a controls adipose tissue inflammation (described in the next sections), miR-30b/c are positive regulators of brown adipogenesis and white adipocyte browning. Both cold exposure and activation of β-adrenergic receptors lead to the upregulation of miR-30b/c, which, via targeting receptor-interacting protein 140 (Rip140), a thermogenic gene co-repressor, promote the thermogenesis and browning of white adipocytes in vitro. Conversely, the knockdown of miR-30b and -30c suppressed the expression of thermogenic genes (including Ucp1) in brown adipocytes both in vitro and in vivo [97]. miR-30b is expressed in human BAT, and WAT, and computational biology methods identified this miRNA as relevant for the development of obesity and T2DM [98,99]. In turn, human obesity is associated with the downregulation of miR-30c in SAT [30].

As in the case of miR-27b, miR-32, miR-155 and miR-455, the data on the role of miR-129-5p in the regulation of white and brown adipogenesis and adipose tissue function are sometimes conflicting. miR-129-5p, on the one hand, by targeting UCP1 inhibitors, namely, insulin growth factor 2 (IGF2) and early growth factor response 1 (EGR1), should enhance thermogenesis in BAT [100]. On the other hand, the content of miR-129-5p was increased in the WAT of the mouse obesity model (db/db mice) and sera of obese patients. Gain- and loss-of-function studies have shown that miR-129-5p inhibits adipocyte differentiation and white adipocyte browning in vitro and decreases the level of specific markers (including UCP1 and PPARγ) in mature white and brown adipocytes [101].

There may be several reasons for such discrepancies. First, the conditions of the in vitro and in vivo experiments may differ from physiological ones. In the case of clinical studies, the selection of study participants with a proper metabolic status is of key importance. It should also be noted that the discrepancy in the results regarding the role of miRNA in the development and thermogenic activity of BAT between rodents and humans may result from the fact that this tissue plays a physiologically different role in the energy balance of small and large mammals. It should also be mentioned that particular miRNA (e.g., miR-23b-5p, miR-135-5p, miR-491-5p and miR-150-3p) are implicated in the regulation of BAT function without affecting brown adipocyte proliferation in pre-clinical studies; however, their significance in the context of human obesity has yet to be determined [102].

## 4. microRNA in the Regulation of Adipose Tissue Function

In addition to being regulators of adipogenesis, adipose tissue browning and thermogenesis, miRNAs have been implicated in other aspects of adipocyte physiology, including lipolysis, lipogenesis and lipid droplet formation, glucose uptake, insulin sensitivity and adipokine secretion. Moreover, they are also involved in the development of obesity-associated adipose tissue dysfunction, manifested as, e.g., metabolic inflammation and insulin resistance. Again, a detailed description of all the processes by which miRNAs are involved in the regulation of the mature adipocyte function is beyond the scope of this review. Here, particular attention will be paid to these aspects of adipocyte physiology that are disturbed in dysfunctional adipose tissue in the course of obesity. 

### 4.1. Lipolysis/Lipogenesis

Fatty acids (FA) are stored in adipocytes in the form of triacylglycerol (TAG) in lipid droplets and mobilized during lipolysis—the catabolic process leading to the breakdown of TAG into glycerol and non-esterified fatty acids (NEFA) for internal or systemic energy use. The basal lipolytic activity of adipocytes is determined by genetic variance, sex, age, physical activity, location of the fat depot, etc., and controlled by multiple factors. Lipolysis is, therefore, a dynamic process involving the assembly and disassembly of protein complexes on the surface of lipid droplets and is regulated by two major opposing hormonal signals, catecholamines and insulin. Since the proteins involved in lipolysis are multifunctional enzymes, lipolysis can mediate homeostatic metabolic signals at the cellular level and participate in interorgan communication. Among the recently identified mediators of lipolysis, such as adipokines, structural membrane proteins, atrial natriuretic peptides, AMPK and mitogen-activated protein kinase (MAPK), are also several miRNAs [103].

An example of miRNAs involved in regulating adipose tissue storage capacity is the miR-181 family, which, as mentioned above, also plays a significant role in the regulation of adipogenesis. On the one hand, the overexpression of miR-181a leads to the downregulation of key lipolytic genes: hormone-sensitive lipase (HSL) and adipose triglyceride lipase (ATGL), leading to the accelerated accumulation of lipid droplets in vitro in porcine primary preadipocytes [57]. On the other hand, miR-181a decreases the expression of genes involved in lipid synthesis and increases the expression of genes involved in β-oxidation via targeting isocitrate dehydrogenase 1 (IDH1, an enzyme in the tricarboxylic acid cycle) and PPARα; therefore, miR-181a transgenic mice exhibit less lipid accumulation as compared with their wild-type littermates [58,104]. This two-directional effect of miR-181a on lipolysis and lipogenesis may partially explain previously described discrepancies regarding the relationship of this miRNA with the development of obesity and its complications in humans [8,34,53,58,59].

miR-143 is another example of a small-noncoding RNA involved both in the regulation of adipogenesis (see previous sections) and lipid metabolism, since its inhibition leads to the downregulation of adipogenic marker genes (including HSL and PPARγ) and subsequent triglyceride accumulation in subcutaneous preadipocytes [40].

Two miRNAs, miR-33a and miR-33b, encoded by the intronic sequences of the genes of sterol regulatory element-binding proteins (SREBPs) 1 and 2, are also implicated in the regulation of cellular cholesterol and fatty acid metabolism [105]. miR-33b, when overexpressed in porcine preadipocytes, via interaction with PPARγ and C/EBPα, attenuates lipid accumulation in vitro [106]. In turn, the genetic ablation of miR-33 in mice leads to enhanced lipid uptake and impaired lipolysis in WAT, which results in the expansion of adipose tissue depots [107]. This effect of miR-33b on adipose tissue can result from its interaction with the HMGA2 (High Mobility Group AT-Hook 2) transcription factor involved in the regulation of preadipocyte proliferation and differentiation [105]. During obesity, both miR-33a and miR-33b are upregulated in VAT in metabolically healthy individuals, while miR-33a is selectively upregulated in patients diagnosed with NAFLD compared to those with NASH [9,53]. This finding is consistent with the fact that the serum miR-33a level is an independent predictor of liver steatosis and inflammation in patients after liver transplantation [108].

miR-425 is another example of an adipocyte lipogenesis and lipolysis regulator. By targeting Cab39, an upstream co-activator of AMPK, miR-425 inhibits intracellular lipolysis and lipid oxidation. This mechanism, together with its ability to interact with mitogen-activated protein kinase 14 (Mapk14) and enhance adipocyte differentiation, determines excessive fat accumulation and the development of obesity in experimental animals with miR-425 overexpression. In turn, miR-425 silencing prevents mice from developing obesity, despite a high-fat diet [109]. Accordingly, in humans, weight loss is associated with the downregulation of miR-425 expression in SAT; however, 3-month lifestyle intervention in T2DM patients was associated with the upregulation of serum miR-425 levels [9,110].

Even though miR-128 exerts a similar effect on preadipocyte differentiation by binding PPARγ, interaction with the SERTA domain containing 2 (Sertad2) can promote lipolysis (measured by HSL and ATGL levels) in 3T3-L1 preadipocytes [111]. The proadipogenic effect seems to be dominant in vivo since human obesity is associated with upregulated miR-128 levels in SAT, which decrease after bariatric surgery [9,37].

ATGL and its co-activator comparative gene identification 58 (CGI-58) are major targets of miR-124a. Subsequently, the ectopic expression of this miRNA in murine preadipocytes leads to reduced lipolysis and increased cellular TAG accumulation [112]. Although the role of miR-124a in the development of human obesity has not been confirmed, it plays a significant role in pancreatic beta cell development and regulation of insulin secretion, and thus its aberrant expression is implicated in the pathogenesis of T2DM [113]. Similarly, miR-145 can inhibit lipolysis in murine preadipocytes by interfering with CGI-58 and forkhead box o1 (FOXO1—another activator of lipolytic activity) [114]. Even though it has not been verified whether gain and loss of function of miR-145 in adipose tissue affect lipolysis and adiposity in vivo, in humans, obesity is associated with its increased expression in SAT, while NAFLD is also associated with its increased expression in VAT [34,37,53]. However, it should be mentioned that the 5p isoform (miR-145-5p) was found to be downregulated in the SAT of morbidly obese individuals, while its serum concentrations are decreased in the course of T2DM and prediabetes [30,115].

As demonstrated, miRNAs, both directly (by interacting with the mRNA of enzymes essential for lipolysis and lipogenesis) and indirectly (by targeting regulators of cellular metabolism, such as PPAR), can influence lipid metabolism and cell lipid storage capacity and thus affect the organism’s ability to accumulate adipose tissue. This action frequently results from their simultaneous effect on adipocyte differentiation and proliferation. 

In addition to being involved in the regulation of adipocyte storage capacity, miRNAs mediate other functions of adipose tissue, including the secretion of adipokines.

### 4.2. Adipokine Secretion

Recent years have changed our perception of adipose tissue from the body’s storage of energy reserves into an active organ that can influence the functioning of other organs and tissues by secreted mediators. The substances secreted by adipose tissue are collectively known as adipokines and can have beneficial and adverse effects on the whole-body function, contributing to obesity-related complications. miRNAs are actively involved in the production and secretion of adipokines in physiology and dysfunctional adipose tissue. However, several adipokines can also regulate miRNA expression.

#### 4.2.1. Adiponectin

Adiponectin, encoded by *ADIPOQ*, is a protein hormone almost exclusively produced in adipose tissue that exhibits favorable metabolic effects, including anti-inflammatory, anti-oxidative and insulin-sensitizing effects. Adiponectin levels measured in the serum and adipose tissue of obese individuals are significantly lower than those in normal-weight subjects and correlate negatively with obesity-related complications [116,117]. There are several lines of evidence that miRNAs are implicated in the obesity-related downregulation of *ADIPOQ* expression. 

miR-378, in addition to being a positive regulator of brown adipogenesis, also plays a role in regulating *ADIPOQ* transcriptional activity. Its overexpression in 3T3-L1 cells leads to a significant decrease in adiponectin mRNA and protein levels, which can be reversed by adding the miRNA-378 inhibitor. Accordingly, miR-378 levels were found to be higher, while adiponectin mRNA levels were found to be lower, in the WAT of diabetic ob/ob mice than wild-type animals [118]. However, its role in the development of human obesity requires further investigation since in human VAT, miR-378 levels correlated positively with *ADIPOQ* expression [119].

In addition to directly targeting *ADIPOQ* mRNA, miRNA can indirectly regulate adiponectin expression by binding key transcription factors. This takes place in the case of miR-144 and FOXO1. miR-144 targets *FoxO1* mRNA, thus reducing its expression and inhibiting its promotional effect on adiponectin, thereby alleviating the inhibitory effect of adiponectin on adipogenesis in porcine preadipocytes [120]. In human obesity, serum miR-144 is a marker of insulin resistance, and its levels are elevated in SAT and decrease after weight loss intervention [9,82,121].

Moreover, miRNAs (e.g., miR-221 and miR-218) play a pivotal role in the post-transcriptional regulation of adiponectin receptors (AdipoR) that mediate adiponectin’s pleiotropic effects in peripheral tissues. Since adiponectin synthesis is reduced in the course of obesity, the induction of AdipoRs via miRNAs could potentially enhance adiponectin’s beneficial effects and ameliorate obesity-associated insulin resistance and diabetes [122,123].

#### 4.2.2. Leptin

Leptin is a 167-amino acid peptide secreted mainly by white adipose tissue, whose levels correlate positively with body adiposity. The pleiotropic functions of leptin include, but are not limited to, the regulation of energy homeostasis, neuroendocrine functions, the immune system, bone metabolism and fertility. Epigenetic changes, including the methylation of regulatory regions, histone modifications and miRNA interference, play a significant role in regulating leptin expression. Furthermore, leptin (and adiponectin) has been observed to epigenetically regulate several miRNAs (reviewed in [124]).

miR-150 downregulates leptin expression by targeting mTOR, while mice with miR-150 knockout have reduced body weight and an elevation in the leptin concentration, in addition to improved glucose tolerance and insulin sensitivity [125]. In turn, miR-155 (an inhibitor of brown adipogenesis and adipose tissue browning) was found to suppress the HFD-induced increase in the plasma leptin concentration in apoE−/− mice [126]. 

At the same time, leptin (together with FFA and resistin) can significantly reduce the expression of MiR-143 in human preadipocytes. Since miR-143 is implicated in the regulation of white adipogenesis and lipolysis, obesity-associated hyperleptinemia may exacerbate adipocyte dysfunction in an autocrine manner [127]. Moreover, leptin can affect the expression of miRNAs in an endocrine manner (e.g., miR-21, miR-27a/b and miR-122) in the liver and, in this way, contribute to obesity-associated liver steatosis (NAFLD) and inflammation (NASH) (reviewed in [124]).

In addition, in obese patients, circulating leptin concentrations have been found to correlate negatively with the expression of miR-325 in VAT and positively with the serum levels of several other miRNAs (e.g., miR-222, miR-143, miR-142-3p, miR-140-5p, miR-130, miR-15a, miR-146a, miR-423-5p, miR-520c-3p and miR-532-5p); however, the pathophysiological links between these findings remain largely unknown [34,55]. 

#### 4.2.3. Other Adipokines

miRNAs are implicated in regulating the expression of several other adipokines and vice versa—adipokines can regulate the expression of miRNAs crucial for adipose tissue development and function. For instance, miR-155, despite being a negative regulator of leptin expression, also downregulates resistin, an adipokine with inverse biological properties to adiponectin, the serum levels of which are elevated in the course of obesity and correlate with the intensity of obesity-associated metabolic inflammation, insulin resistance, NAFLD, atherosclerosis and cardiovascular diseases. Subsequently, miR-155-/- mice are characterized by increased resistin expression in white adipose tissues [128]. 

Visfatin (an adipokine that reduces glucose release from liver cells and stimulates glucose utilization in adipocytes and myocytes by binding the insulin receptor) can target miR-34a and miR-181a—two miRNAs crucial for the development and function of BAT and WAT, respectively [129]. In turn, omentin (an adipokine secreted mainly by VAT, which is responsible for maintaining insulin sensitivity, exerting anti-inflammatory, anti-atherosclerotic and cardiovascular protective effects) was shown to downregulate miR-27a with the subsequent inhibition of oxidative stress and inflammation [130].

Apart from modifying the inflammatory milieu via adipokines, miRNAs also have a direct impact on pro-inflammatory responses in adipose tissue by regulating cytokine expression.

### 4.3. Inflammation

Chronic overnutrition, manifested by the excessive accumulation of lipids, impairs adipocyte metabolism, leading to mitochondrial dysfunction that contributes to endoplasmic reticulum stress, hypoxia and cell hypertrophy. These pathological changes activate the expression of genes encoding cytokines, chemokines and adhesion molecules in adipose tissue, which attracts infiltrating immune cells (different subsets of T cells and macrophages) that contribute to the production of pro-inflammatory cytokines. Pro-inflammatory mediators (e.g., tumor necrosis factor-alpha, TNFα and interleukins (IL) 1 and 6) impair adipose tissue function in an auto- and paracrine manner but also influence other tissues, contributing to the development of insulin resistance and other components of metabolic syndrome [131]. There is mounting evidence that this chronic, low-grade inflammation, called metaflammation, is under the epigenetic control of miRNAs, and conversely, the inflammation of the adipose tissue leads to the dysregulation of miRNA expression. Accordingly, the miRNA panel of adipose tissue in genetically obese ob/ob mice resembles that of TNFα-treated 3T3-L1 preadipocytes, suggesting that obesity leads to the spontaneous conversion of the miRNA profile to a pro-inflammatory one [69].

The effects of miRNAs on the inflammatory response in adipose tissue can be twofold: stimulating and inhibiting. For instance, the overexpression of miR-132 in primary human adipose-derived stem cells leads to an increase in the production of IL8 and monocyte chemoattractant protein-1 (MCP1), while the overexpression of miR-126 in human adipocyte progenitor cells leads to the downregulation of MCP1 [132,133]. In turn, the exposure of human differentiated adipocytes to miR-145, miR-26a and miR-let-7d results in the downregulation of TNFα synthesis in the case of miR-26a and let-7d, and upregulation in the case of miR-145 [134]. 

miR-30a is an example of an miRNA triggering anti-inflammatory responses in adipose tissue. By targeting the signal transducer and activator of transcription 1 (*STAT1*), miR-30a opposes the actions of interferon γ (IFNγ), resulting in increased insulin sensitivity in DIO mice [135]. Moreover, members of the MiR-30 family are involved in the polarization of macrophages towards the M2 (anti-inflammatory) phenotype. HFD causes the hypermethylation of MiR-30 genes via the activation of AMPK and delta-like ligand 4 (DLL4)-Notch signaling, leading to their downregulation and the exacerbation of inflammation and insulin resistance in an animal model of obesity [136]. Human obesity is associated with decreased miR-30a expression in SAT but with elevated serum levels [30,137].

miR-17 (by blocking *STAT3* and apoptosis signal-regulating kinase 1—*ASK1* expression) can also suppress pro-inflammatory responses [138,139]. Its overexpression reduces the secretion of IL1β, IL6 and TNFα in lipopolysaccharide-stimulated macrophages, preventing macrophage-mediated adipose tissue inflammation and improving insulin resistance [139]. Notably, miR-17 levels are reduced in the VAT and sera of obese individuals [140]. 

In turn, the overexpression of miR-27a enhances the polarization of macrophages towards a pro-inflammatory phenotype (M1) by targeting PPARγ. Conversely, miR-27a knockout reduces cytokine (e.g., IL10) expression in activated macrophages [141]. Moreover, HFD leads to increased MiR-27a serum levels, which correlates with increased adiposity and insulin resistance. However, in miR-27a knockout animals, cytokine levels are within the normal range, despite being fed an HFD [142].

The pro-inflammatory environment in adipose tissue contributes to the dysregulation of adipogenesis, since pro-inflammatory cytokines (e.g., TNFα) can downregulate the expression of key adipogenic factors, e.g., *PPARG*, *C/EBPA* and *FABP4*. There is evidence that miRNA participates in this process: for instance, TNFα, via activation of the nuclear factor κB pathway, induces the expression of miR-130 (an inhibitor of adipocyte differentiation) in murine adipocytes [143]. Moreover, HFD triggers miR-130b activation in adipose tissue, which, via targeting PPARγ, polarizes macrophages to express the M1 phenotype, exacerbating inflammation and insulin resistance [144].

Moreover, the influence of pro-inflammatory cytokines on miRNA expression in adipocytes can be miRNA specific. While the treatment of human mature adipocytes with TNFα and IL6 leads to a significant elevation in miR-335 expression, in a culture of adipose tissue-derived MSCs from obese subjects, the expression of miR-221 correlates negatively with *TNFA* mRNA levels [145,146].

### 4.4. Insulin Resistance

Insulin resistance is determined by impaired sensitivity to insulin in its main target organs: adipose tissue, liver and muscle. In properly functioning adipose tissue, insulin decreases lipolysis, thereby reducing FFA efflux from adipocytes. However, in adipose tissue dysfunction, the action of insulin is disturbed, leading to increased circulating FFA concentrations and ectopic fat accumulation. The pathogenesis of obesity-associated insulin resistance is multifactorial, and all the abovementioned phenomena (impaired lipolysis, adipokine secretion and pro-inflammatory state) contribute to its development. There is also evidence that miRNA can directly impair insulin signaling in adipose tissue. 

miR-143, in addition to its role in regulating adipogenesis and lipolysis, remains an independent risk factor for insulin resistance. By targeting insulin-like growth factor 2 receptor (IGF2R), miR-143 impairs insulin signaling in 3T3-L1 preadipocytes, while its inhibition protects mice on an HFD against the development of obesity-associated insulin resistance. Accordingly, in obese individuals with metabolic syndrome, circulating levels of miR-143-3p are significantly elevated compared to metabolically healthy controls [147].

Similarly, two other regulators of adipocyte differentiation, miR-103 and miR-107, which decrease the expression of caveolin-1 (Cav-1)—an essential mediator of insulin signaling—can diminish the number of insulin receptors and impair downstream insulin signaling. Their silencing results in the upregulation of Cav-1 expression in the adipose and liver tissues of DIO mice, which translates to improved insulin signaling and glucose uptake and implicates these miRNAs as novel therapeutic targets for the treatment of metabolic syndrome [148]. Targeting insulin signaling pathways also plays a role in the modulation of insulin sensitivity via miR-146 and miR-30d. By upregulating secreted frizzled-related protein 4 (SFRP4)—an adipokine which increases insulin resistance by decreasing the abundance of insulin receptor substrate 1 (IRS1) and the FOXO1 transcription factor—these two miRNAs can aggravate insulin resistance in the course of human obesity [149,150].

Another way in which miRNAs may regulate insulin sensitivity in adipose tissue is through their influence on the expression of genes encoding glucose transporters (GLUTs). For instance, miR-181a suppression decreases the expression of glucose transporters 1 and 4 (GLUT1 and GLUT4) in porcine preadipocytes [57]. Activation of GLUT4 signaling (via the suppression of the PH domain and Leucine-rich repeat Protein Phosphatase 2—PHLPP2) also plays a role in the insulin-sensitizing effect of miR-130a in 3T3-L1 cells and mice on an HFD [151].

miRNAs involved in the regulation of adipose tissue development and function and dysregulated in the course of human obesity are summarized in Table 1 and Table 2.

## 5. Final Remarks and Conclusions

Currently, the main aim of obesity treatment is not only to reduce body weight but also to reduce the risk of associated complications, which contribute to the deterioration of quality of life. Therefore, there is a continuous need for research on new obesity treatments that, in addition to weight loss, help reduce the risk of related complications.

Could miRNAs be considered as new drugs for obesity? It is too early to answer this question affirmatively. Taking into account their pleiotropic actions in adipose tissue, which include the control of adipogenesis, adipose tissue browning and critical processes (e.g., lipolysis and adipokine synthesis) in adipose tissue physiology, as well as obesity-related dysfunction (e.g., pro-inflammatory activity and insulin resistance), miRNAs constitute a valuable target for new therapeutic approaches. However, the pleiotropic and multidirectional action of miRNAs is a severe limitation of their application for treating human diseases and prevents their systemic administration. In addition, the most promising results regarding the role of miRNAs in obesity treatment are derived from in vitro and animal studies and do not allow us to assume that similar effects will be obtained in humans.

On the other hand, a promising direction of research on miRNAs is evaluating their role as markers of disease and organ dysfunction. MiRNAs released from adipose tissue are now considered a new category of adipokines, and their profile evolves with the development of obesity. There has been a search for miRNAs that could constitute early predictors of the development of, e.g., NAFLD and its progression to NASH or insulin resistance, and the search for those that could predict its progression to T2DM is underway. Identifying such marker miRNAs in sera would allow for the early implementation of interventions aiming to prevent the development of obesity-related complications. Based on the available literature, at the moment, elevated serum levels of miR-34a, miR-103/miR-107, miR-143-3p, miR-144 and miR-378 can be considered predictors of insulin resistance, while miR-193b can be considered a predictor of prediabetes [70,77,82,121,148]. Therefore, individuals with high serum concentrations of these miRNAs may benefit from lifestyle interventions based on a low-glycemic index diet. In turn, a low circulating miR-155 level may suggest a need for regular control of the lipid profile [93]. On the contrary, high miR-145-5p and miR-199a-5p serum concentrations seem to be a marker of metabolic health [8,115]. Blood levels of some other miRNAs (e.g., miR-425) can also act as predictors of successful lifestyle interventions in obese diabetic patients [110].

## Figures and Tables

**Table 1 cells-10-03342-t001:** MicroRNA involved in regulation of white adipose tissue development and function.

MicroRNA	Target	Function	Experimental Model	Reference	Human Obesity and Related Complications			Reference
					VAT	SAT	Serum	
miR-9-5p	Wnt3a	↑ adipogenic differentiation	rat MSCs	[31]	↑ in obesity		↑ in diabetic nephropathy	[9] [32]
miR-210	TCF7L2	↑ adipogenic differentiation	3T3-L1 murine preadipocytes	[33]		↑ in obesity ↓ after weight loss ↑ in obese metabolically healthy individuals	↑ in diabetic retinopathy	[9] [34,35]
miR-21	TGFβ1	↑ adipogenesis	human ADSCs	[36]		↑ in obesity ↓ after weight loss	↓ in obesity	[9] [37,38,39]
miR-199a-5p	TGFβ/Smad	↑ adipogenesis	human BMSCs	[40]	↑ in obesity	↓ in obese metabolically healthy individuals	↑ in obesity	[8] [34] [41]
miR-143	ERK5	↓ adipogenic differentiation during clonal expansion	rat ADSCs	[43]		↓ after weight loss		[9]
		↑ adipogenic differentiation during growth arrest					↑ in obesity	[44]
	HSL PPARγ IGF2R	↑ triglyceride accumulation ↑ insulin resistance	human preadipocytes 3T3-L1 murine preadipocytes HFD obese mice	[42] [147]			↑ in metabolic syndrome and insulin resistance	[147]
miR-204	RUNX2	↑ adipogenic differentiation	human BMSCs	[45]	↑ in obesity	↑ in obesity		[9] [37]
miR-637	OSX	↑ adipogenic differentiation	human MSCs	[46]			↑ weight loss	[47]
miR-27b	PPARγ	↓ adipogenic differentiation	human BMSCs	[48]	↓ in obese individuals with NASH	↓ in obese individuals with T2DM ↑ in obese metabolically healthy individuals	↓ in obese individuals with PCOS	[9] [30] [37] [53,54]
miR-130a	PPARγ	↓ adipogenic differentiation	human BMSCs	[48]		↓↑ in obesity	↑ in morbidly obese adolescents	[30] [37] [55]
miR-31	C/EBPα	↓ adipogenic differentiation	human ADSC	[50]	↑ in obesity		↑ in obese adolescents	[9] [56]
miR-181a	PPARγ TNFα IDH1 PHLPP2	↑ adipogenic differentiation ↓ lipolysis ↑ insulin sensitivity	porcine primary preadipocytes 3T3-L1 murine preadipocytes mice on HFD	[57] [151]		↑ in obesity ↑ in obese individuals with T2DM		[8] [34]
	PPARα	↑ lipid accumulation	hepatocytes	[58]			↑ in obese individuals with NAFLD	[53]
		↓ lipid synthesis ↓ lipid accumulation	MEF mice on HFD	[104]	↓ in obese individuals with NASH	↓ in obesity		[58,59]
miR-146b	SIRT1/PPARγ SFRP4	↑ adipogenic differentiation ↑ insulin resistance	3T3-L1 murine preadipocytes insulin-resistant mice	[60] [150]	↓ in obese individuals with NASH and pericellular fibrosis	↑ in obesity	↑ in obesity predictor of T2DM risk	[9] [53] [61,62] [149]
miR-103	RAI14 AKT/mTOR MEFD2	↑ adipogenic differentiation	porcine preadipocytes 3T3-L1 murine preadipocytes	[65,66]	=in obesity	=in obesity		[67]
miR-103/miR-107	ATF6 Cav-1	↑ adipocyte apoptosis ↑ insulin resistance	primary murine preadipocytes DIO mice	[68] [148]	↓ in obese individuals with NASH	↓ in obesity ↓ after weight loss	↑ in obesity	[9] [37] [53] [69] [70]
miR-33a/b	PPARγ C/EBPα HMGA2	↓ lipid accumulation ↓ adipogenic differentiation and proliferation	porcine preadipocytes human preadipocytes miR-33 -/- mice	[105,106,107]	↑ in obesity ↑ in obese individuals with NAFLD		↑ in individuals with NAFLD after liver transplantation	[9] [53] [108]
miR-425	Mapk14 Cab39	↑ adipogenic differentiation ↓ adipogenic proliferation ↓ lipolysis ↓ lipid oxidation	3T3-L1 murine preadipocytes miR-425 -/- mice	[109]		↑ in obesity ↓ after weight loss	↑ after lifestyle intervention in T2DM	[9] [110]
miR-128	PPARγ Sertad2	↓ adipogenic differentiation and proliferation ↑ lipolysis	3T3-L1 murine preadipocytes	[111]		↑ in the course of obesity ↓ after weight loss		[9] [37]
miR-124a	ATGL CGI-58	↓ lipolysis	OP9 preadipocytes murine primary adipocytes	[112]				
miR-145	FOXO1 CGI-58	↓ lipolysis	murine primary adipocytes 3T3-L1 preadipocytes	[114]	↑ in obese individuals with NAFLD	↑ in obesity ↓ in the course of obesity (*5p)	↓ in T2DM patients (*5p)	[30] [34] [37] [53] [115]
miR-144	FOXO1	↓ adiponectin synthesis ↑ adipogenic proliferation	porcine preadipocytes	[120]		↑ in obesity ↓ after weight loss	↑ in insulin resistance	[9] [82] [121]
miR-30a	STAT1	↓ inflammation ↑ insulin sensitivity ↑ M2 macrophages	DIO mice	[135]		↑ in obesity	↑ in obesity	[8] [137]
miR-17	STAT3 ASK1	↓ IL1β, IL6 and TNFα	myeloid-derived suppressor cells LPS stimulated macrophages	[138] [139]	↓ in obesity		↓ in obesity	[140]

↓ downregulation; ↑ upregulation; ADSC—adipose tissue-derived stromal cell; AKT/mTOR—protein kinase B/mammalian target of rapamycin; ASK1—apoptosis signal-regulating kinase 1; ATF6—activating transcription factor 6; ATGL—adipose triglyceride lipase; BMSCs—human bone marrow stromal cells; Cav-1—caveolin-1; Cab39—calcium-binding protein 39; C/EBPα—CCAAT/enhancer-binding protein α; CGI-58—comparative gene identification 58; DIO—diet-induced obesity; ERK5—extracellular-signal-regulated kinase 5; FOXO1—forkhead box o1; HFD—high-fat diet; HMGA2—high mobility group AT-hook 2; HSL—hormone sensitive lipase; IDH1—isocitrate dehydrogenase 1; IGF2R—insulin-like growth factor 2 receptor; IL—interleukin; LPS—lipopolysaccharide; Mapk14—mitogen-activated protein kinase 14; MEF—mouse embryonic fibroblasts; MEFD2—myocyte enhancer factor 2D; MSC—mesenchymal stem cells; NAFLD—non-alcoholic fatty liver disease; NASH—non-alcoholic steatohepatitis; OSX—osterix; PCOS—polycystic ovary syndrome; PHLPP2—PH domain and leucine-rich repeat protein phosphatase 2; PPARα—peroxisome-proliferator-activated receptor α; PPARγ—peroxisome-proliferator-activated receptor γ; RAI14—retinoic acid-induced protein 14; RUNX2—runt-related transcription factor 2; SAT—subcutaneous adipose tissue; Sertad2—SERTA domain containing 2; SFRP4—secreted frizzled-related protein 4; SIRT1—sirtuin 1; STAT1—signal transducer and activator of transcription 1; STAT3—signal transducer and activator of transcription 3; TCF7L2—T cell-specific transcription factor 7-like 2; TGFβ—transforming growth factor β; TNFα—tumor necrosis factor-alpha; T2DM—type 2 diabetes mellitus; VAT—visceral adipose tissue; Wnt3a—a Wnt ligand.

**Table 2 cells-10-03342-t002:** MicroRNA involved in regulation of brown adipose tissue development and function.

MicroRNA	Target	Function	Experimental Model	Reference	Human Obesity and Related Complications			Reference
					VAT	SAT	Serum	
**Constitutive brown adipose tissue (cBAT)**								
miR-193a/b miR-365 cluster	RUNX1T1 Cdo IGFBP5	↑ brown adipogenesis ↓ myogenesis =brown adipogenesis	primary murine brown preadipocytes *miR-193b*-mutant mice	[74,75]	↓ in obese individuals with NASH	↓in obesity ↑ after bariatric surgery	↓in obesity ↑ in prediabetes ↓ upon exercise	[8,9] [30] [53] [76,77]
miR-328	BACE1	↑ brown adipogenesis ↓ myogenesis ↓ weight gain	primary murine brown preadipocytes DIO mice	[78]	↓ in obese individuals with NASH		↓in obesity	[53] [79,80]
miR-378	PDE1B	↑ brown adipogenesis	ob/ob mice DIO mice	[81]	↓ in obese individuals with NASH	↓in obesity	↑ in insulin resistance	[30] [53] [82]
	adiponectin	↓ adiponectin synthesis	3T3-L1 cells ob/ob mice	[118]	↓in obesity			[119]
miR-203 miR-182	UCP1 PGC1α PPARα Cox7 Cox8	↑ brown adipogenesis ↑ thermogenesis	Dgcr8 -/- mice	[83]	↑in obesity	↓after weight loss		[9] [34]
miR-106b miR-93 cluster	UCP1 PRMD16 PPARα PGC1α	↓ brown adipogenesis ↓ thermogenesis	DIO mice	[84]	↓↑in obesity	↓after weight loss		[9] [30]
**Recruitable brown adipose tissue (rBAT)**								
miR-34a	FGF21 SIRT1	↓ brown adipogenesis ↓adipocyte browning ↓ thermogenesis	3T3-L1 adipocytes DIO mice	[64]	↑in obesity	↑in obesity	↑in insulin resistance	[8] [37] [59] [82]
miR-27b	PPARγ PPARα C/EBPα Prohibitin UCP1 PRMD16, PGC1α	↓ brown adipogenesis ↓adipocyte browning ↓ mitochondrial biogenesis ↓ thermogenesis	human MADS	[86,87,88]	↓ in obese individuals with NASH	↓in obesity		[30] [37] [53]
miR-133	PPARγ PPARα PRMD16, PGC1α	↓ brown adipogenesis ↓adipocyte browning ↓ thermogenesis	murine primary adipocytes C57Bl/6N mice	[90,91]	↑in obesity			[9]
miR-155	C/EBPβ	↓ brown adipogenesis ↓adipocyte browning ↓ thermogenesis ↓ leptin synthesis ↓ resistin synthesis	murine primary adipocytes C57BL/6J mice apoE−/− mice miR-155-/- mice	[6] [92] [126] [128]		↓in diabetic obese individuals	correlates negatively with BMI	[34] [93]
miR-32	Tob1	↑ brown adipogenesis ↑ adipocyte browning ↑ thermogenesis	murine brown preadipocyte WT-1 cells murine primary adipocytes C57BL/6J mice	[7]		↓after weight-loss	=in obesity	[9] [82]
miR-455	RUNX1T1 Necdin HIF1AN	↑ brown adipogenesis ↑ adipocyte browning ↑ thermogenesis	murine brown preadipocytes C3H10T1/2 cells 3T3-L1 cells C57BL/6 mice	[94,95,96]		↑in obesity		[8]
miR-30b/c	Rip140	↑ adipocyte browning ↑ thermogenesis	murine white and brown preadipocytes C57BL/6 mice	[97]		↓in obesity		[30]
miR-129-5p	IGF2 EGR1 FABP4 UCP1 PPARγ	↓ adipogenesis ↓adipocyte browning	murine primary adipocytes db/db mice	[101]			↑in obesity	[101]

↓ downregulation; ↑ upregulation; apoE—apolipoprotein E; Bace1—β-secretase 1; BMI—body mass index; Cdo—surface protein Cdo; C/EBPα—CCAAT/enhancer-binding protein α; C/EBPβ—CCAAT/enhancer-binding protein β; Cox7—cytochrome c oxidase subunit 7; Cox8—cytochrome c oxidase subunit 7; db/db mice—mice homozygous for the diabetes spontaneous mutation; Dgcr8—DiGeorge syndrome critical region 8; DIO—diet-induced obesity; EGR1—early growth factor response 1; FABP4—fatty acid binding protein 4; FGF21—fibroblast growth factor 21; HIF1AN—hypoxia inducible factor 1 subunit alpha inhibitor; IGFBP5—insulin-like growth factor binding protein 5; IGF2—insulin growth factor 2; MADS—multipotent adipose-derived stem cells; ob/ob mice—mice homozygous for the obese spontaneous mutation; PDE1B—phosphodiesterase 1B; PGC1α—peroxisome PPARγ coactivator 1 α; PPARα—peroxisome-proliferator-activated receptor α; PPARγ—peroxisome-proliferator-activated receptor γ; PRMD16—PR domain containing 16; Rip140—receptor-interacting protein 140; RUNXT1T—runt-related transcription factor 1 partner transcriptional co-repressor 1; SAT—subcutaneous adipose tissue; SIRT1—sirtuin 1; Tob1—transducer of erbB2 1; VAT—visceral adipose tissue; UCP1—uncoupling protein 1.

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
