# Peer review of "microRNAs in Human Adipose Tissue Physiology and Dysfunction"

_cells, 2021, doi:10.3390/cells10123342_

Round 1

Reviewer 1 Report

This is a solid review of the literature surrounding  to the role of microRNAs in the physiology of adipose tissue. It is well documented, but there are a number of changes that the author should make.
The number and order of the sections that the author uses in the review is confusing. In the first place it puts the Introduction section, in second place it puts miRNAs as regulators of gene expression and third the role of microRNAs in adipogenesis, and goes directly to section 2.1. miRNA in the regulation of white adipogenesis. Section 3 is later renamed microRNA in adipose tissue function. The same happens at the end of line 584 where you would have to start section 4.
It would be beneficial to the reader if the authors could provide more precise conclusions from the data they have studied.

Author Response

Reviewer 1

This is a solid review of the literature surrounding  to the role of microRNAs in the physiology of adipose tissue. It is well documented, but there are a number of changes that the author should make.

The number and order of the sections that the author uses in the review is confusing. In the first place it puts the Introduction section, in second place it puts miRNAs as regulators of gene expression and third the role of microRNAs in adipogenesis, and goes directly to section 2.1. miRNA in the regulation of white adipogenesis. Section 3 is later renamed microRNA in adipose tissue function. The same happens at the end of line 584 where you would have to start section 4.

It would be beneficial to the reader if the authors could provide more precise conclusions from the data they have studied.

I would like to express my gratitude to the Reviewer for the positive reception of the manuscript and for drawing attention to some weak points I tried to improve.

  • The number and order of the sections that the author uses in the review is confusing. In the first place it puts the Introduction section, in second place it puts miRNAs as regulators of gene expression and third the role of microRNAs in adipogenesis, and goes directly to section 2.1. miRNA in the regulation of white adipogenesis. Section 3 is later renamed microRNA in adipose tissue function. The same happens at the end of line 584 where you would have to start section 4.

I agree with the Reviewer that in the initial version of the manuscript numbering of the sections and their order was confusing. To clarify the manuscript concept, a new paragraph summarizing its structure was added to the Introduction section. Moreover, each of the following sections has been supplemented with a summary that constitutes an introduction to the next part of the work.

“It begins with a brief overview of the mechanisms by which miRNAs may regulate gene expression to provide a background for discussion of their role in adipose tissue development and function. Then, the role of miRNA in white and brown adipogenesis and the browning of white adipose tissue is presented. Finally, the role of various miRNAs in regulating the physiological functions of adipose tissue (including lipolysis/lipogenesis and adipokine secretion) and pathological states (inflammatory activity and insulin resistance) is discussed." (Lines 65-71)

“Discovery of the mechanisms by which miRNAs can influence gene expression has led to an understanding of their role in regulating the development and function of many organs and tissues, including adipose tissue.” (Lines 145-147)

“For many years, BAT has been thought to play a marginal role in adult energy homeostasis. However, recent research has increased our understanding of the mechanisms involved in the development and activation of brown adipocytes and their contribution to metabolic health. It has occurred that miRNAs also play a significant role in regulating these processes.” (Lines 303-307)

“Since miRNAs involved in brown adipogenesis frequently participate also in regulation of thermogenic pathways, they will be discussed together in the following sections.” (Lines 328-330)

"In addition to being regulators of adipogenesis, adipose tissue browning, and thermogenesis, miRNAs have been implicated in other aspects of adipocyte physiology, including lipolysis, lipogenesis and lipids droplet formation, glucose uptake, insulin sensitivity, and adipokine secretion. Moreover, they are also involved in the development of obesity-associated adipose tissue dysfunction, manifested as, e.g., metabolic inflammation and insulin resistance." (Lines 512-517)

“In addition to being involved in the regulation of adipocyte storage capacity, miRNAs mediate other functions of adipose tissue, including the secretion of adipokines.” (Lines 602-603)

“Apart from modifying inflammatory milieu via adipokines, miRNAs have also a direct impact on pro-inflammatory responses in adipose tissue by regulating cytokines expression.” (Lines 683-685)

In addition, the confusion regarding the manuscript structure could have resulted from careless editing. In the initial version, the serial numbers of the chapters and paragraphs were unordered. Some of these mistakes resulted from my neglect, but some from the automatic formatting when sending the document. Therefore, I have made every effort to improve the revised version of the manuscript in terms of editing. 

  • It would be beneficial to the reader if the authors could provide more precise conclusions from the data they have studied.

Following the Reviewer's suggestion, a novel paragraph summarizing the main findings on the potential application of miRNAs in diagnosing and preventing obesity-related complications has been added to the Final remarks and conclusions section.

“Based on the available literature, at the moment, elevated serum levels of miR-34a, miR-103/miR-107, miR-143-3p, miR-144, and miR-378 can be considered are predictors of insulin resistance, while miR-193b – of prediabetes [70,77,82,121,148]. Therefore individuals with high serum concentrations of these miRNAs may benefit from lifestyle interventions based on a low glycemic index diet. In turn, a low circulating miR-155 level may suggest a need for regular control of lipid profile [93]. On the contrary, high miR-145-5p and miR-199a-5p serum concentrations seem to be a marker of metabolic health [8,115]. A blood level of some other miRNA (e.g., miR-425) can also act as a predictor of the successful lifestyle intervention in obese diabetic patients [110]." (Lines 837-846)

Reviewer 2 Report

Authors bring to the readers a comprehensive revision about genes, pathways, miRNAs and regulation patterns of different aspects related to adipogenesis and adipose function. I found the revision quite interesting and useful. 

Author Response

Reviewer 2

Authors bring to the readers a comprehensive revision about genes, pathways, miRNAs and regulation patterns of different aspects related to adipogenesis and adipose function. I found the revision quite interesting and useful. 

I would like to express my gratitude to the Reviewer for the positive reception of the manuscript.

Reviewer 3 Report

In this manuscript, they reviewed the role of microRNA in adipose tissue physiology and dysfunction. In adipocyte physiology, they focused on adipocyte differentiation, proliferation, and browning. Then they discussed the microRNA in adipocyte functions, including lipolysis, lipogenesis, adipokine, and so on. They reviewed the recent articles and represented them in an accurate and objective view. Overall, this article is worth publishing. Only a few minor comments:

  1. The serial numbers of the chapters and paragraphs are unordered after Line 140.
  2. In Table1, they listed two targets of miR-17 but included only the reference of ASK1.

Author Response

Reviewer 3

In this manuscript, they reviewed the role of microRNA in adipose tissue physiology and dysfunction. In adipocyte physiology, they focused on adipocyte differentiation, proliferation, and browning. Then they discussed the microRNA in adipocyte functions, including lipolysis, lipogenesis, adipokine, and so on. They reviewed the recent articles and represented them in an accurate and objective view. Overall, this article is worth publishing. Only a few minor comments:

I would like to express my gratitude to the Reviewer for the positive reception of the manuscript and for drawing attention to some weak points I tried to improve.

  1. The serial numbers of the chapters and paragraphs are unordered after Line 140.

I have to apologize for the careless editing of the manuscript. Some of the mistakes result from my neglect, but some from the automatic formatting when sending the document. I have made every effort to improve the revised version of the manuscript in terms of editing. 

  1. In Table1, they listed two targets of miR-17 but included only the reference of ASK1.

I thank the Reviewer again for drawing attention to this important issue. A proper reference was added in Table 1 and in the main text.

139.

Zhang, M.; Liu, Q.; Mi, S.; Liang, X.; Zhang, Z.; Su, X.; Liu, J.; Chen, Y.; Wang, M.; Zhang, Y.; Guo, F.; Zhang, Z.; Yang, R.Both miR-17-5p and miR-20a alleviate suppressive potential of myeloid-derived suppressor cells by modulating STAT3 expression. J Immunol 2011, 186, 4716–4724.